# Effects of Creatine Supplementation on Athletic Performance in Soccer Players: A Systematic Review and Meta-Analysis

**DOI:** 10.3390/nu11040757

**Published:** 2019-03-31

**Authors:** Juan Mielgo-Ayuso, Julio Calleja-Gonzalez, Diego Marqués-Jiménez, Alberto Caballero-García, Alfredo Córdova, Diego Fernández-Lázaro

**Affiliations:** 1Department of Biochemistry and Physiology, School of Physical Therapy, University of Valladolid, 42004 Soria, Spain; a.cordova@bio.uva.es; 2Laboratory of Human Performance, Department of Physical Education and Sport, Faculty of Education, University of the Basque Country, 01007 Vitoria, Spain; julio.calleja.gonzalez@gmail.com; 3Academy Department, Deportivo Alavés SAD, 01007 Vitoria-Gasteiz, Spain; diegomarquesjimenez@gmail.com; 4Department of Anatomy and Radiology, Faculty of Physical Therapy, University of Valladolid, Campus de Soria, 42004 Soria, Spain; albcab@ah.uva.es; 5Department of Cellular Biology, Histology and Pharmacology, Faculty of Physical Therapy, University of Valladolid, Campus de Soria, 42004 Soria, Spain; diego.fernandez.lazaro@uva.es

**Keywords:** sport nutrition, nutritional supplements, recovery, team sports, physical performance, ergogenic aids

## Abstract

Studies have shown that creatine supplementation increases intramuscular creatine concentrations, favoring the energy system of phosphagens, which may help explain the observed improvements in high-intensity exercise performance. However, research on physical performance in soccer has shown controversial results, in part because the energy system used is not taken into account. The main aim of this investigation was to perform a systematic review and meta-analysis to determine the efficacy of creatine supplementation for increasing performance in skills related to soccer depending upon the type of metabolism used (aerobic, phosphagen, and anaerobic metabolism). A structured search was carried out following the Preferred Reporting Items for Systematic Review and Meta-Analyses (PRISMA) guidelines in the Medline/PubMed and Web of Science, Cochrane Library, and Scopus databases until January 2019. The search included studies with a double-blind and randomized experimental design in which creatine supplementation was compared to an identical placebo situation (dose, duration, timing, and drug appearance). There were no filters applied to the soccer players’ level, gender, or age. A final meta-analysis was performed using the random effects model and pooled standardized mean differences (SMD) (Hedges’s *g*). Nine studies published were included in the meta-analysis. This revealed that creatine supplementation did not present beneficial effects on aerobic performance tests (SMD, −0.05; 95% confidence interval (CI), −0.37 to 0.28; *p* = 0.78) and phosphagen metabolism performance tests (strength, single jump, single sprint, and agility tests: SMD, 0.21; 95% CI, −0.03 to 0.45; *p* = 0.08). However, creatine supplementation showed beneficial effects on anaerobic performance tests (SMD, 1.23; 95% CI, 0.55–1.91; *p* <0.001). Concretely, creatine demonstrated a large and significant effect on Wingate test performance (SMD, 2.26; 95% CI, 1.40–3.11; *p* <0.001). In conclusion, creatine supplementation with a loading dose of 20–30 g/day, divided 3–4 times per day, ingested for 6 to 7 days, and followed by 5 g/day for 9 weeks or with a low dose of 3 mg/kg/day for 14 days presents positive effects on improving physical performance tests related to anaerobic metabolism, especially anaerobic power, in soccer players.

## 1. Introduction

Creatine (Cr) is one of the most used ergogenic nutritional aids by athletes. Studies have shown that the effective dose for Cr supplementation of 0.3 g/kg/day for 5–7 days, followed by a maintenance dose of 0.03 g/kg/day (most commonly for 4–6 weeks) [1], increases intramuscular phosphocreatine (PCr) concentrations by favoring phosphagen metabolism [2]. This may help explain the observed improvements in the performance of high-intensity exercises that lead to greater training adaptations [3,4]. Likewise, Cr supplementation has been shown to increase the glycogen replenishment rate, which may help those athletes who perform at prolonged submaximal effort (65–75% peak of the maximum rate of oxygen consumption − VO_2max_) [5] or engage in repeated high-intensity exercises [6,7], in relation to aerobic and anaerobic metabolism, respectively. In this line, there are many sports modalities, the practice of which involves a combination of high-intensity actions, in isolation or repeatedly, where optimal anaerobic metabolism is necessary, and low-intensity actions, where efficient aerobic metabolism is necessary [8].

Among them, in particular, soccer is characterized by combining high-intensity activities, such as sprinting, running, jogging, accelerating, jumping, and changing direction, with low-intensity phases (stopping or walking) [9]. The average distance covered by players during a soccer match is between 8 and 12 km [10], where they perform between 50 and 250 high-intensity actions [11] that represent about 1–12% of the total distance covered [12]. In this sense, during the high- and maximum-intensity phases, the energy that a player gets is obtained through anaerobic processes (both phosphagen and anaerobic metabolism), whereas during the general effort of a soccer match (90 or 120 min), it is obtained aerobically [13]. Therefore, having substrates that provide the necessary energy in each phase seems to be an objective to obtain maximal performance [14].

There have been numerous studies focusing on the effects of Cr supplementation on physical soccer performance, with mixed results. Thus, while Biwer et al. [15] and Williams et al. [16] showed possible benefits for aerobic performance, other authors did not find these benefits [17,18,19]. Similar results have been observed regarding tests that use phosphagen metabolism. In this case, while there are several studies that have observed potential positive effects of Cr supplementation on this type of metabolism [6,18,20], Ramírez-Campillo et al. [19] and Williams et al. [16] did not find any benefits for the performance of individual actions, such as a single jump or a sprint. However, all of the studies that have focused on the effect of Cr supplementation in relation to anaerobic metabolism, such as repeated sprints [17,19,21] or the Wingate test [6,7], showed possible beneficial effects, although the final effect on soccer athletic performance is unknown, given that some of these benefits are rather small [17,19,21].

Although Cr could improve soccer athletic performance, to the best of the authors’ knowledge, there is no clear consensus on the kind of soccer skills, and therefore the energy system involved, for which Cr supplementation could be more effective. Moreover, there is some controversy regarding the doses, duration, and timing. Hence, unifying the data of these different studies would support the soccer world by making it possible to apply this knowledge over the course of a season. Therefore, we proposed carrying out a systematic review and final meta-analysis of the relevant articles published in the scientific literature, the main aim of which is to discern the potential effects of Cr on soccer athletic performance depending on the metabolic energy system used (aerobic, phosphagen, and anaerobic metabolism). Thus, this systematic review and meta-analysis presents current information on the effects of Cr on soccer athletic performance. In addition, it shows the effective doses and ideal moment of its intake.

## 2. Methods

### 2.1. Searching Strategies

The present article is a systematic review with a meta-analysis focusing on the effect of Cr or Cr monohydrate on soccer performance. It was carried out following the Preferred Reporting Items for Systematic Review and Meta-Analyses (PRISMA) guidelines, which helped to improve the integrity of this review [22]. The PICOS model was used to determine the inclusion criteria [23]—P (Population): “soccer players”, I (Intervention): “creatine supplementation”, C (Comparators): “same conditions with placebo”, O (Outcome): “soccer-specific skills and relationships with aerobic, phosphagen, and anaerobic metabolism performance”, and S (study design): “double-blind and randomized design”.

A structured search was conducted in the following databases: PubMed/MEDLINE, web of science (WOS), Cochrane Library, and Scopus. It included results until 30 January 2019, while no year restriction was applied to the search strategy. Search terms included a mix of medical subject headings (MeSH) and free-text words for key concepts related to Cr and soccer performance. Specifically, we used the following search equation: (“football” [All Fields] OR “soccer” [All Fields]) AND “creatine supplementation” [All Fields] AND (“physical performance” [All Fields] OR “physical endurance” [All Fields] OR “physical” [All Fields] OR “endurance” [All Fields] OR “performance” [All Fields] OR “aerobic” [All Fields] OR “anaerobic” [All Fields]), which returned relevant articles in the field of applying the snowball strategy. All titles and abstracts from the search were cross-referenced to identify duplicates and any potential missing studies. Titles and abstracts were screened for a subsequent full-text review. The search for published studies was independently performed by two different authors (JMA and JCG) and disagreements were resolved through discussions between them.

### 2.2. Inclusion and Exclusion Criteria

There were no filters applied to the soccer players’ level, gender, race, or age to increase the power of the analysis. However, for the articles obtained in the database search, the following inclusion criteria were applied to select the final studies: (1) in which there was an experimental condition that included the ingestion of Cr before and/or during exercise which was compared to an identical experimental condition with the ingestion of a placebo; (2) testing the effects of Cr on soccer-specific tests and/or real or simulated matches; (3) with a blinded and randomized design; (4) with clear information regarding the administration of Cr (relative dose of Cr per kilogram of body mass and/or absolute dose of Cr with information about body mass, timing of Cr intake before the onset of performance measurements, etc.); (4) on soccer players with previous training backgrounds in this sport; and (5) published in any language. On the other hand, the following exclusion criteria were applied to the experimental protocols of the investigation: (1) studies that were not conducted with soccer players; (2) studies that were performed for clinical purposes or therapeutic use; (3) the absence of a true placebo condition; and (4) studies carried out using participants with a previous medical condition, illness, or injury.

### 2.3. Data Extraction

Once the inclusion/exclusion criteria were applied to each study, data on study source (including authors and year of publication), study design, Cr supplementation (dose and timing), sample size, characteristics of the participants (level and gender), and final outcomes of the interventions were extracted independently by two authors using a spreadsheet. Subsequently, disagreements were resolved through discussion until a consensus was achieved.

Experiments were clustered by the type of test used to assess team sport performance, and groups of experiments were created which assessed the effect of Cr on aerobic performance (Yo-Yo intermittent recovery test level 1), phosphagen metabolism performance (strength, jump, sprint, and agility course), and anaerobic metabolism measures (repeated sprint ability and the Wingate test). Six studies included measurements of two or more types of performance outcomes (e.g., aerobic and phosphagen abilities) or even two types of tests for the same performance outcome. In these cases, each test or type of performance outcome was treated as a single and independent set of data for the meta-analysis and included in the appropriate performance outcome.

The mean (M), standard deviation (SD), and sample size data were extracted by one author from the tables of all of the included papers (DMJ). Whenever necessary, we contacted the authors to obtain the data. When it was impossible, mean and SD were extrapolated from the figures. Any disagreement was resolved by consensus (JMA and DMJ) or third-party adjudication (JCG).

### 2.4. Quality Assessment of the Experiments

Methodological quality and risk of bias were assessed by two authors independently (JMA and DMJ), and disagreements were resolved by third-party evaluation (JCG), in accordance with the Cochrane Collaboration Guidelines [24]. The items on the list were divided into different domains: random sequence generation (selection bias), allocation concealment (selection bias), blinding of participants and personnel (performance bias), blinding of outcome assessment (detection bias), incomplete outcome data (attrition bias), selective reporting (reporting bias), and other types of bias. They were characterized as “low” if criteria for a low risk of bias were met (plausible bias unlikely to seriously alter the results) or “high” if criteria for a high risk of bias were met (plausible bias that seriously weakens confidence in the results). If the risk of bias was unknown, it was considered “unclear” (plausible bias that raises some doubts about the results). Full details are given in Table 1 and Figure 1.

### 2.5. Statistical Analysis

Descriptive data of the participants’ characteristics are reported as mean ± standard deviation. Descriptive analyses and figures of risk of bias were performed using a spreadsheet (Microsoft Excel 2016^©^ USA), whereas meta-analytic statistics were made with Review Manager (RevMan) version 5.3 (Copenhagen: The Nordic Cochrane Centre, The Cochrane Collaboration, 2014). The standardized mean difference (SMD), the number of participants, and the standard error of the SMD for each study were used to quantify changes in the performance variables when comparing the ingestion of Cr vs. a placebo. SMDs for each study group were calculated using Hedges’s *g* [25]. SMDs were weighted by the inverse of variance to calculate an overall effect and its 95% confidence interval (CI). The net treatment effect was obtained by subtracting the SMD of the control group from the SMD of the experimental group. Variance was calculated from the pooled SD of change scores in both groups. Considering that the effect of Cr on performance may differ according to dose and other moderators relating to participants, we decided to use a random effects model with the DerSimonian and Laird method [26]. The Cohen criteria were used to interpret the magnitude of SMD (MSMD): <0.2, trivial; 0.2–0.5, small; 0.5–0.8, moderate; and >0.8, large [27].

To avoid problems using Q statistic to assess systematic differences (heterogeneity), we calculated the I^2^ statistic, which indicated the percentage of observed total variation across studies that was due to real heterogeneity rather than chance [24]. I^2^ interpretation is intuitive and lies between 0% and 100%. An I^2^ value between 25% and 50% represents a small amount of inconsistency, an I^2^ value between 50% and 75% represents a medium amount of heterogeneity, and an I^2^ value >75% represents a large amount of heterogeneity [28]. A restrictive categorization of values for I^2^ would not be appropriate for all circumstances, although it would tentatively accept adjectives of low, moderate, and high to I^2^ values of 25%, 50%, and 75%, respectively [28,29,30].

## 3. Results

### 3.1. Main Search

The literature search identified a total of 101 articles related to the selected descriptors, but only nine articles met all the inclusion criteria (see Figure 2). From these 101 articles, 19 of them were removed because they were duplicative. From the remaining 82 articles screened, 11 papers were removed because they were narrative or systematic reviews and another 19 did not match the search descriptions. From the 52 full-text articles assessed for eligibility, another 43 papers were removed because they were unrelated to the effects of Cr on soccer physical performance. The topics and number of studies that were excluded were: 1 article about gene polymorphisms; 7 articles that combined Cr with other supplements, such as beta-alanine and beta-hydroxy-beta-methylbutyrate (HMB); 20 articles on the effects of Cr on different sports, such as American football and rugby; 1 related to muscle damage; 1 analyzing biochemical markers; 10 on clinical markers; and 3 about the self-reported prevalence of Cr consumption. Thus, the current systematic review and meta-analysis included nine studies.

### 3.2. Creatine Supplementation

The participant and intervention characteristics of the experiments included in this systematic review are depicted in Table 2, whereas the summary of studies included is shown in Table 3. The total sample consisted of 168 soccer players (118 males, 50 females) with an age of 20.3 ± 2.0 years (from 15 to 30 years, as an average for the experimental sample).

In this context, Table 3 shows the samples included in all studies, which consisted of players who competed at levels from professional or elite (*n* = 2) to semiprofessional or amateur teams (*n* = 7), who had well-established training habits and whose age group varied from under-17 (*n* = 2) to senior team categories (*n* = 7). In two out of nine studies, Cr was administered based on an individual’s body mass, while an absolute dose was provided for all participants in seven studies. In this way, the doses used in each study included values of 30 g/day (3 × 10 g) in one study, 20 g/day (4 × 5 g) in six studies, 0.3 g/kg, four times in one day in one study, and 0.03 g/day in one study. In five of the studies included, the time of ingestion of Cr was along with a meal (breakfast, lunch, and dinner) or separated by 3–4 h, while the others (*n* = 4) did not mention the time of ingestion. Further, there were three out of nine studies that used a loading dose of 20 g/day (divided into four doses) for a week and then a dose of 5 g/day for periods of 1 day, 5 weeks, and 6 weeks, respectively. On the other hand, there were four studies in which the durations were 6 days, 2 of 7 days, 1 of 14 days, 1 of 6 weeks, and 1 of 7 weeks, respectively.

### 3.3. Effect on Aerobic Performance Meta-Analysis

Figure 3 shows the overall effect of Cr supplementation on aerobic performance and indicates that Cr did not produce any significant effect on aerobic performance (SMD, −0.05; 95% CI, −0.37 to 0.28; MSMD, trivial; I^2^, 0%; *p* = 0.78). Only Biwer et al. [15] on a submaximal treadmill run interspersed with high intensity, and Williams et al. [16] on ball-sport endurance and speed test mean circuit time (s) (aerobic), presented improvements in favor of Cr supplementation.

### 3.4. Effect on Phosphagen Metabolism Performance Meta-Analysis

Pre-exercise Cr ingestion produced small but not significant increases in physical performance in tests mainly related to phosphagen metabolism performance (SMD, 0.21; 95% CI, −0.03 to 0.45; MSMD; small; I^2^,43%; *p* = 0.08) (Figure 4). The results indicated that Cr is associated with moderate but not significant improvements in strength performance (one-repetition maximum (1 RM), peak torque) (SMD, 0.50; 95% CI, −0.15 to 1.14; MSMD, moderate; I^2^,72%; *p* = 0.13). Concretely, Bemben et al. [6] showed improvements in favor of Cr on neuromuscular strength (1 RM) for bench press, squat, and power clean; isokinetic strength (peak torque) for quadriceps (180°/s and 300°/s); and isokinetic strength (peak torque) for quadriceps and hamstrings (300°/s). Likewise, the results presented trivial and not significant improvements in single jump performance (SMD, 0.14; 95% CI, −0.12 to 0.39; MSMD, trivial; I^2^, 0%; *p* = 0.28). In favor of Cr supplementation, Mujika et al. [17] showed improvements on a recovery counter-movement jump (CMJ) consisting of three jumps (average data); Ostojic [18] on a vertical jump (cm); Claudino et al. [20] on a CMJ; and Ramírez-Campillo et al. [19] on peak jump power load (kg), squat jump (cm), and drop jump (40 cm) reactive strength index (mm/ms). Similarly, the results showed trivial and not significant improvements in single sprint velocity SMD, 0.06; 95% CI, −0.70 to 0.81); MSMD, trivial; I^2^, 62%; *p* = 0.88). Only Ostojic [18] displayed improvements in favor of Cr supplementation on a sprint power test (s). Likewise, the results showed trivial and not significant improvements in the time required to complete agility tests (SMD, −0.11; 95% CI, −0.83 to 0.61; MSMD, trivial; I^2^, 0%; *p* = 0.77). Only Cox et al. [21] presented improvements in favor of Cr supplementation on a simulated match play test/agility run.

### 3.5. Effect on Anaerobic Performance Meta-Analysis

However, a large and significant, potentially ergogenic effect of Cr was found in those tests which were mainly related to anaerobic performance (SMD, 1.23; 95% CI 0.55–1.91; MSMD, large; I^2^, 81%; *p* <0.001) (Figure 5). Cr supplementation demonstrated a large and significant effect on the Wingate test (SMD, 2.26; 95% CI, 1.40–3.11; MSMD, large; I^2^, 72%; *p* <0.001). Also, Bemben et al. [6] showed an improvement in anaerobic power and capacity measured by the Wingate test, and Yañez-Silva et al. [7] presented improvements in favor of Cr supplementation in peak and mean power output (W/k) and total work (J/kg) measured by the Wingate test. On the other hand, the results showed small but not significant effects on repeated spring ability performance (SMD, 0.26; 95% CI –0.13 to 0.65; MSMD, trivial; I^2^, 0%; *p* = 0.20). Mujika et al. [17] presented an improvement in favor of Cr supplementation on a repeated sprint test consisting of six maximal 15-m runs with a 30-s recovery, Cox et al. [21] on a simulated match play test/20-m repeated sprint time, and Ramírez-Campillo et al. [19] on a running anaerobic sprint test.

## 4. Discussion

The main purpose of this systematic review and meta-analysis was to summarize the effects of Cr supplementation on physical performance tests related to aerobic, phosphagen, and anaerobic metabolisms in soccer players.

The main results indicate that Cr supplementation with 20–30 g/day ingested for 6–7 days followed by 5 g/day for 1–9 weeks led to significant improvements in anaerobic performance. On the other hand, Cr supplementation showed trivial to small but not significant effects on tests related to aerobic and phosphagen metabolisms. Thus, this meta-analysis suggests that Cr supplementation could be an ergogenic aid to improve the anaerobic performance of soccer players.

The best method of increasing muscle Cr stores appears to be Cr supplementation with a loading phase of 0.3 g/kg/day (~20–30g/day) for 3–5 days, followed by 3–5 g/day to maintain elevated Cr stores [31]. Likewise, Cr supplementation with 0.03 g/kg/day (~2–3 g/day) will increase muscle Cr stores over a 3–4 weeks period [31]. These protocols are important since Cr levels in the human body can be elevated for up to 30 days [31,32]. Likewise, it has to be taken into to account that about 20–30% of individuals do not respond to Cr loading [33]. However, the articles included in this review and meta-analysis did not contain any comments about responders or non-responders when soccer players were supplemented with Cr. Therefore, although the studies that were included in this review and meta-analysis complied with the protocols aimed at increasing muscle Cr stores, to our knowledge, it is impossible to know if the results were affected by players who were responders or non-responders to Cr loading.

The physiological demands of soccer have changed dramatically over time [34]. Currently, soccer players cover greater distances, perform more explosive actions, and compete at higher intensities than ever before [35,36], and sports science has played a key role in these advances [37]. In particular, nutrition has been integral in the search for and use of supplements that allow players to perform better at higher intensities [8]. In this sense, Cr is one of the most studied supplements in athletes, given that it produces the ability to resynthesize the adenosine triphosphate (ATP) that is used while exercising and, consequently, to maintain maximal exercise increases [31]. Thus, Cr supplementation could lead to greater training adaptations due to the higher quality of and capacity for exercise, as well as a quicker recovery period [38]. For this reason, Cr could be recommended for the improvement of soccer physical performance because it is involved in different metabolic pathways [39].

### 4.1. Effect on Aerobic Performance

Soccer requires a great aerobic capacity because of the duration of a match (90 min and sometimes 30 min of extra time) [40]. In addition, a relationship between the aerobic power and competitive classification, the competitive level of the team, and the distance covered during the match has been demonstrated [41,42]. For this reason, looking for ways to improve this capacity throughout the season is essential to maintaining a high level of performance. In this sense, oral supplementation with Cr for 7 days could improve aerobic performance in elite athletes [43], since it has been shown that after Cr supplementation, there is an increase in PCr content at rest in muscle fiber type I [44]. In addition to a reduction in the use of muscle PCr, there is a lower accumulation of inorganic phosphate (Pi) as well as a decrease in muscle pH during low-intensity exercises after loading Cr, which would indicate a delay of fatigue during prolonged periods of resistance work [45]. However, these results are controversial, since in most studies, Cr supplementation did not improve the ability to perform long-term submaximal exercise [46,47], nor did it modify the maximum absorption of oxygen and the circulatory, metabolic, and ventilatory responses to a progressive exercise test [48,49]. In this line, Mujika et al. [17], Ostojic [18], and Biwer et al. [15] did not observe changes in aerobic capacity in both male and female soccer players and young soccer players after supplementation with 20 and 30 g/day of Cr for 6–7 days. Further, Ramirez et al. did not find improvements in aerobic performance after a 6-week supplementation with Cr in female amateur players [19]. Therefore, although aerobic metabolism plays a major role in soccer, given that it provides 90% of the energy used during soccer match play [50], the results obtained in this meta-analysis indicate that Cr supplementation in soccer players has no benefits in improving aerobic performance. Highly trained aerobic metabolism is dependent on intramuscular triglycerides and not PCr or muscle glycogen [51]. For that reason, Cr supplementation could not enhance aerobic performance [52].

### 4.2. Effect on Phosphagen Metabolism Performance

Anaerobic power and explosive force are also essential components of soccer performance since they allow players to execute the constant muscular adjustments necessary to perform different actions, as well as allowing actions such as jumps, shots, short sprints, or agility actions [53]. In this sense, the power of the lower extremities as a product of neuromuscular stimulus has been associated with speed performance in soccer players [54]. Thus, several studies have examined the potential associations among the ability to sprint and several measures of strength and power in different exercises related to soccer performance [54,55]. It seems reasonable to expect soccer players to benefit from Cr supplementation because most of their activity in this field involves powerful and explosive anaerobic movements that require the immediate release of energy provided by ATP and the rapid re-synthesis of ATP from adenosine diphosphate (ADP) and PCr. In this way, the results obtained in the meta-analysis indicated that the intake of Cr prior to exercise was associated with a small but not significant increase in physical performance in those tests that were mainly related to phosphagen metabolism performance. Specifically, the results showed moderate but not significant effects on different strength exercises (1 RM and peak torque) performed by 25 college football players after 6 days of supplementation with 20 g/day of Cr [6]. However, only trivial and nonsignificant improvements were seen in single jump performance, single sprint speed, or the time required to complete agility tests. Although this type of activity requires ATP, its re-synthesis is not decisive for jumping or sprinting because they are more dependent on neuromuscular performance [56]. In addition, the short duration of the Cr supplementation protocols used (5 g, four times) in this meta-analysis could also have influenced in the results.

### 4.3. Effect on Anaerobic Performance

It is currently recognized that the most decisive actions during soccer practice are related to anaerobic metabolism [57]. In this sense, anaerobic power, together with the specific skills of the sport, seem to be the determinants of high performance [58]. Although there are different tests to assess anaerobic power and, therefore, the performance of the anaerobic metabolism of a player, the most important test used in the field of soccer exercise physiology is the gold standard anaerobic Wingate test [59,60]. For that reason, the Wingate test has been used to validate field tests in this sport [61,62] because it has been positively correlated with better performance during soccer matches [63]. In addition, it has been used to monitor the effectiveness of different training programs [64,65]. Therefore, seeking better adaptations in anaerobic metabolism seems to favor the performance of soccer players. In this sense, Cr seems to be a good ergogenic aid to produce improvements in test performances in which anaerobic metabolism predominates, as it has been demonstrated in this meta-analysis. Specifically, both Yañez-Silva et al. [7] and Bemben et al. [6] showed that short-term Cr supplementation (6–7 days) improved maximum and average anaerobic power, as well as the total work measured by the Wingate test in both young and university players. The positive effect of Cr supplementation on activities related to anaerobic metabolism may be due to the benefits that Cr has on the muscle glycogen store [5]. This effect is thought to be the result of increased cell size due to Cr-induced water retention and is associated with the upregulation of signaling pathways mediating glycogen and protein synthesis, namely, 5’AMP-activated protein kinase (AMPK) and mechanistic target of rapamycin (mTOR)-mediated signaling [38].

### 4.4. Strengths, Limitations, and Future Lines of Research

The main limitation of this systematic review and meta-analysis is the scarcity of studies carried out in relation to Cr supplementation in soccer players (*n* = 9), which forced us to carry out the analyses by mixing data of both sexes, different competitive levels, and different research protocols. Thus, it should be noted that neither the dose nor the duration of Cr supplementation protocol has been taken into account. In fact, studies were mixed, in which Cr supplementation used short-term (5–7 days) and long-term (6–7 weeks) protocols, which may have influenced the results. The protocols used in some studies [16,21] may have also influenced the results obtained in this meta-analysis because some physical performance tests were joined together as a field test simulating match play (not isolated), which probably could affect the metabolic pathway of the participants during exercise and, consequently, the performance obtained in each test. However, one important strength is that the proportion of the total variation that was attributable to the heterogeneity observed in many of the physical tests analyzed was zero (I^2^ = 0%). Another limitation of this systematic review and meta-analysis is related to the impossibility of presenting body composition data that would have helped us interpret the results.

Future research projects could include the use of a general protocol of Cr supplementation in different skills related to soccer performance that involve different metabolisms in both male and female soccer players, as well as in different levels of competition.

### 4.5. Practical Applications

Generally, Cr supplementation has been used during the competitive season (in case of fatigue) in order to sustain adequate levels of Cr, PCr, and/or glycogen in the muscles and also to improve muscle repair [66]. However, Cr supplementation could be specially used in those periods in which the priority of training is to improve anaerobic power or enhance muscular power and adaptation, such as during pre-season, winter break, or a summer-like opportunity window [67]. In addition, Cr supplementation could be particularly useful for those team members who, due to their position in the field and the match characteristics, are specially involved in actions that demand high anaerobic power. Related to this point, it should be kept in mind that some individuals may respond more (or less) to Cr supplementation than others because they have lower endogenous muscle Cr content and/or a greater amount of type II fibers [66]. Consequently, effects should be evaluated and/or measured individually in each soccer player, interrupting supplementation in athletes who do not respond to it (likely “low-responders”) or report different effects than expected, given that about 20–30% of individuals do not respond to Cr loading [33]. Finally, but no less importantly, soccer players could also benefit from Cr supplementation to augment their training capacity (increase in load and/or performance) when they return to normal training after an injury, which could prevent reinjury due to the physical demands of soccer training or competition. Regardless, future studies should analyze the individual and optimal load for each player based on personalized needs [68].

## 5. Conclusions

The results of this systematic review and meta-analysis have shown that Cr supplementation improved the performance of physical tests related to anaerobic metabolism, especially anaerobic power, in soccer players. The effective dose of Cr supplementation to obtain positive effects should include a load dose of 20–30 g/day, divided 3–4 times a day, ingested for 6–7 days, and followed by 5 g/day for 9 weeks or a low dose of 3 mg/kg/day for 14 days or more.

## Figures and Tables

**Figure 1 nutrients-11-00757-f001:**
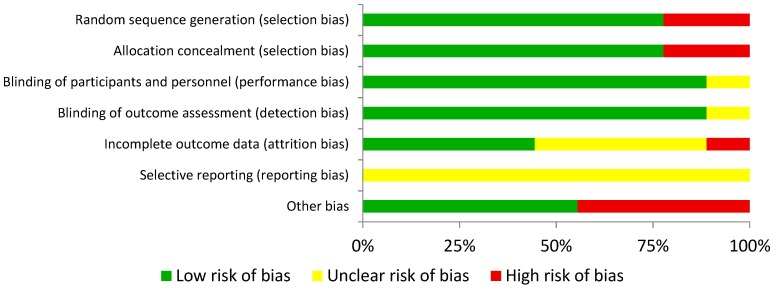
Risk of bias summary: review authors’ judgements about each risk of bias item for each included study.

**Figure 2 nutrients-11-00757-f002:**
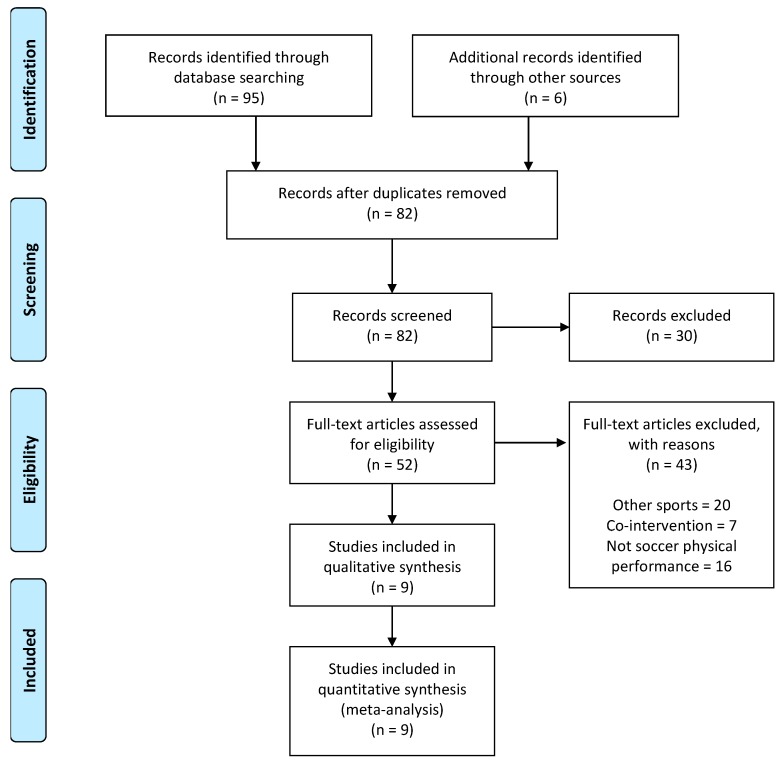
Selection of studies.

**Figure 3 nutrients-11-00757-f003:**
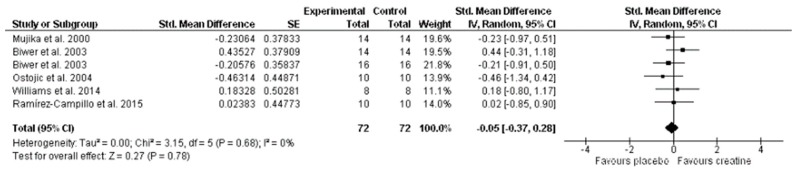
Forest plot comparing the effects of creatine supplementation on aerobic performance.

**Figure 4 nutrients-11-00757-f004:**
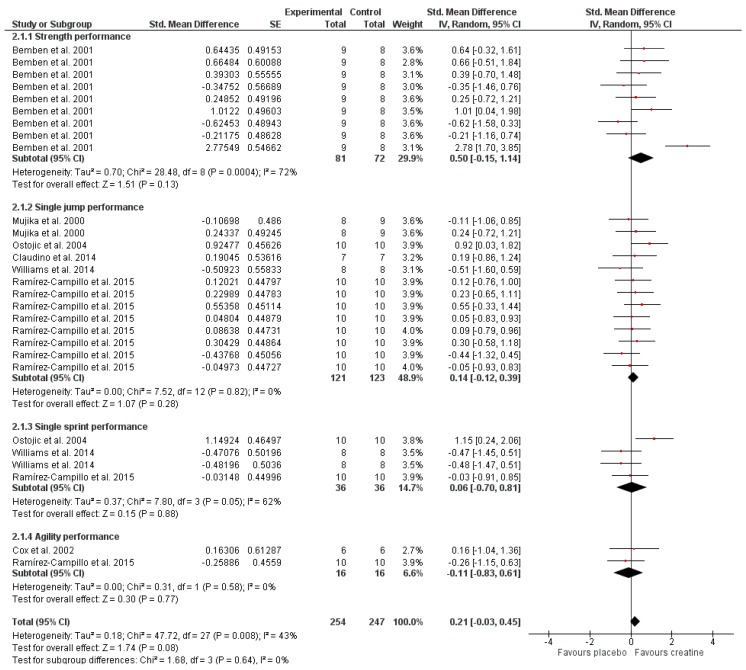
Forest plot comparing the effects of creatine supplementation on phosphagen metabolism performance.

**Figure 5 nutrients-11-00757-f005:**
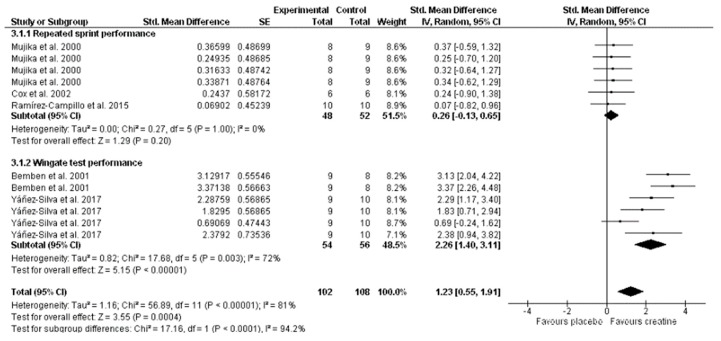
Forest plot comparing the effects of creatine supplementation on anaerobic performance.

**Table 1 nutrients-11-00757-t001:** Risk of bias graph: review of authors’ judgements about each risk of bias item presented as percentages across all included studies. 

 indicates low risk of bias, 
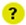
 indicates unknown risk of bias, and 
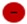
 indicates high risk of bias.

	Random sequence generation (selection bias)	Allocation concealment (selection bias)	Blinding of participants and personnel (performance bias)	Blinding of outcome assessment (detection bias)	Incomplete outcome data (attrition bias)	Selective reporting (reporting bias)	Other bias
Mujika et al., 2000 [17]					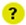	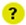	
Bemben et al., 2001 [6]		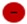				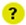	
Cox et al., 2002 [21]	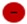	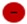			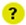	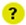	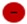
Biwer et al., 2003 [15]						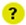	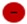
Ostojic 2004 [18]			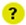	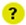		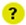	
Claudino et al., 2014 [20]					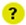	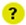	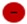
Williams et al., 2014 [16]	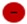					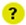	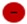
Ramírez-Campillo et al., 2015 [19]					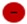	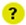	
Yáñez-Silva et al., 2017 [7]					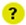	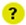	

**Table 2 nutrients-11-00757-t002:** Participant and intervention characteristics of the studies included in the systematic review. Cr: creatine.

Level of participants	Elite	2 studies [17,21]
	Semiprofessional or amateur	7 studies [6,7,15,16,18,19,20]
Age group of participants	Under-17	2 studies [7,18]
	Senior	7 studies [6,15,16,17,19,20,21]
Type of Cr administration	Based on individual’s body mass	2 studies [7,15]
	Absolute dose	7 studies [6,16,17,18,19,20,21]
Dose used	30 g/day (3 doses of 10 g daily)	1 study [18]
	20 g/day (4 doses of 5 g daily)	6 studies [6,16,17,19,20,21]
	0.3 g/kg, four times in one day	1 study [15]
	0.03 g/kg/day	1 study [7]
Time of ingestion	Along with breakfast/lunch/dinner or separated by 3–4 h	5 studies [6,15,16,19,20]
	Not mentioned	4 studies [16,17,18,21]
Loading phase	20 g/day (in 4 doses) for a week plus 1 dose of 5 g/day for 9 weeks	1 study [6]
	20 g/day (in 4 doses) for a week plus 1 dose of 5 g/day for 5 weeks	1 study [19]
	20 g/day (in 4 doses) for a week plus 1 dose of 5 g/day for 6 weeks	1 study [20]
	No loading phase	6 studies [7,15,16,17,18,21]
Duration of treatment	6 days	3 studies [15,17,21]
	7 days	2 studies [16,18]
	14 days	1 study [7]
	6 weeks	1 study [19]
	7 weeks	1 study [20]
	9 weeks	1 study [6]

**Table 3 nutrients-11-00757-t003:** Summary of studies included in the systematic review.

AUTHOR/S- YEAR	POPULATION	INTERVENTION	OUTCOMES ANALYZED	MAIN CONCLUSIONS
Mujika et al., 2000 [17]	17 highly trained male players (20.3 ± 1.4 years)	5 g, 4 times/day for 6 days	Countermovement jump Repeat sprint ability Intermittent endurance test	↔ ↑ ↔
Bemben et al., 2001 [6]	25 male university players (19.3 ± 0.5 years)	5 g, 4 times/day (separated by 3–4 h) for 5 days 5 g/day for 9 weeks	Neuromuscular strength tests Anaerobic power test Isokinetic test	↑ ↑ ↔
Cox et al., 2002 [21]	12 elite female players (22.1 ± 5.4 years)	5 g, 4 times a day for 6 days	Sprint test Agility racing test Agility kick drill test	↑ ↑ ↔
Biwer et al., 2003 [15]	15 (7 males and 8 females) university players Age not presented	0.3 g/kg, 4 times/day (after breakfast, lunch, and dinner and before bedtime) for 6 days	Submaximal running test	↔
Ostojic, 2004 [18]	20 young male players (16.6 ± 1.9 years)	10 g, 3 times/day for 7 days	Dribbling test Spring test Endurance test Countermovement jump	↑ ↑ ↔ ↑
Claudino et al., 2014 [20]	14 male professional players (18.3 ± 0.9 years)	5 g, 4 times/day (breakfast, lunch, dinner, and before bedtime) for 7 days 5 g/day for 6 weeks	Countermovement jump	↑
Williams et al., 2014 [16]	16 amateur male players (26.0 ± 4.5 years)	5 g, 4 times/day (~4 h between doses) for 7 days	Aerobic (circuit time) Speed (12- and 20-m sprint) Explosive power (vertical jump)	↔ ↔ ↔
Ramírez-Campillo et al., 2015 [19]	30 amateur female players (22.9 ± 2.5 years)	5 g, 4 times/day (at breakfast, lunch, dinner, and before bedtime) for 7 days 5 g/day (at lunch) for 5 weeks	Jump test Repeated sprint test Resistance Speed performance in direction change	↑ ↑ ↔ ↑
Yañez-Siva et al., 2017 [7]	19 young male players (17.0 ± 0.5 years)	0.03 g/kg/day (at midday meal) for 14 days	Maximal power test Average output power test Fatigue index test Total work test	↑ ↑ ↔ ↑

↑: statistically significant increase; ↔ change with no statistical significance; ↓: statistically significant decrease.

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
