# Peer review of "Effects of Creatine Supplementation on Athletic Performance in Soccer Players: A Systematic Review and Meta-Analysis"

_nutrients, 2019, doi:10.3390/nu11040757_

Round 1
Reviewer 1 Report
Better use of lists or tables would help readability in a number of places, for example, the number of studies demonstrating improvements (line 221-28)(again 233-36).
Lactic and alactic are not common terms, strongly prefer phosphagen and anaerobic systems.
We have known for about 15-20 years that creatine might improve anaerobic activities by about 2-3% and in the anaerobic systems, this paper just demonstrates it in soccer by examining soccer-specific papers.
Any analysis on responders vs non-responders as part of this paper?
The paper would benefit from why you are partitioning the effects to the three different energy systems. Clinically does this change how you would use the creatine or how you would change the athlete's training routine? In other words, answer the "so what" in application.
Author Response
The authors appreciate the time you devoted to reading our manuscript and helping us to craft an improved version. We are pleased to clarify your concerns which we believe have improved the quality and applicability of your work. Please, find below our response to your observations. We have made a concerted attempt to systematically address the specific concerns raised for this revision and we have highlighted the alterations to this revision within the manuscript in green for your convenience.
Reviewer(s)' Comments to Author:
Better use of lists or tables would help readability in a number of places, for example, the number of studies demonstrating improvements (line 221-28)(again 233-36).
Answer: Thank you for your suggestion. The authors have modified some places of results to help readability. In this line, the authors have added table 2 where summarize studies included in this review with meta-analysis (see 3.2 section). Moreover, the authors have included some information about studies demonstrating improvements in 3.3, 3.4 and 3.5 sections (please see these sections).
Lactic and alactic are not common terms, strongly prefer phosphagen and anaerobic systems.
Answer: Thank you for your recommendation. The authors have changed alactic by phosphagen metabolism and have deleted lactic, using only anaerobic metabolism.
We have known for about 15-20 years that creatine might improve anaerobic activities by about 2-3% and in the anaerobic systems, this paper just demonstrates it in soccer by examining soccer-specific papers.
Answer: Thank you for your comment. The authors agree with the reviewer. For many years it has been known that creatine could favor actions that specially use the phosphagen metabolism given its impact on the synthesis of ATP through PCr. However, there are many studies that despite this knowledge have continued to determine the supplementation with creatine in activities that use another type of energetic metabolism. Thus, in soccer, one of the most practiced sports in the world and with the greatest economic impact, research has continued to be carried out to find out in which type of activities its supplementation is most effective. With this review with meta-analysis it could be shown that creatine supplementation is only effective to improve anaerobic power and not so much in those actions that require phosphagen systems to obtain energy as one might think.
Any analysis on responders vs non-responders as part of this paper?
Answer: Thank you for that suggestion. It is a very interesting topic. The Creatine supplementation of 20 g per day for at least 3 days has resulted in significant increases in total Cr for some individuals but not others, suggesting that there are 'responders' and 'non-responders'. In this line, the authors have added next paragraph: “Likewise, it has to take into to account that about 20-30% of individuals do not respond to Cr loading [33]. However, the articles included in this review and meta-analysis didn’t described any comment about responders or non-responders when the soccer players were supplemented with Cr. Therefore, although the studies that were included in this review and meta-analysis complied with the protocols aimed at increasing the muscle Cr stores, for the authors knowledge, it is impossible to know if the results of them were affected by the players were responders or non-responders to Cr loading.”
The paper would benefit from why you are partitioning the effects to the three different energy systems. Clinically does this change how you would use the creatine or how you would change the athlete's training routine? In other words, answer the "so what" in application.
Answer: Thank you for your recommendation. The authors have added a new paragraph where explain the practical applications of the conclusions. This paragraph is:
4.5. Practical applications
Generally, Cr supplementation has been used during the competitive season (in case of fatigue), in order to sustain adequate levels of CR, PCr and/or glycogen in the muscles or also to improve the muscles’ repair [66]. But Cr supplementation could be specially used in those periods in which the priority of training is to improve anaerobic power or enhance their muscular power and adaptation, such as pre-season, winter-break or on summer like opportunity window [67]. In addition, Cr supplementation could be really useful for those members of the team who, due to their position in the field and the match characteristics, are specially involved in actions that demand a high anaerobic power. Related to this point, it should be kept in mind that some individuals may respond more (or less) to Cr supplementation than others because they have lower endogenous muscle Cr content and/or greater population of type II fibers [66]. Consequently, effects should be evaluated and/or measured individually in each soccer player, interrupting the supplementation in athletes who do not respond to it (likely ‘‘low-responders’’) or who report different effects than expected given that about 20-30%of individuals do not respond to Cr loading [33]. Finally, but not less important, soccer players could also benefit from it in order to augment their training capacity (increase in load and/or performance) when they return to train normally after an injury, preventing them for re-injury due to the physical demands of soccer training or competition. Anyway, future lines will analyze the individual and optimal load for each player based on personalized needs.”

Reviewer 2 Report
All the comments are included in the text; so please refer to the text.

Author Response
The authors appreciate the time you devoted to reading our manuscript and helping us to craft an improved version. We are pleased to clarify your concerns which we believe have improved the quality and applicability of your work. Please, find below our response to your observations. We have made a concerted attempt to systematically address the specific concerns raised for this revision and we have highlighted the alterations to this revision within the manuscript in green for your convenience.
Reviewer(s)' Comments to Author:
Line 18-20: The opening sentence is a bit unclear so I would suggest re-phrasing it.
Answer: Thank you for your suggestion. The authors have re-phrased these lines as: “Studies have shown that creatine supplementation increases intramuscular creatine concentrations, favoring the energy system of phosphagens, which may help explain the observed improvements in high intensity exercise performance. However, research on soccer physical performance has shown controversial results in part because the energy system used has not been taken into account.”
Line 27: What does that mean? Identical in terms of what? Taste, form, color
Answer: Thank you for your interest. The authors have included into brackets (the doses, duration, timing and drug appearance) to explain what means identical.
Line 38: I would recommend providing the exact time range here as thic could be misleading
Answer: Thank you for your recommendation. Taking into account that the improvements have been shown in 2 different protocols, the authors have re-written abstract conclusion as: “In conclusion, creatine supplementation with a high dose of 20-30 g/day divided 3-4 times per day ingested for 6-7 days followed by 5 g/day until 9 weeks or with a low dose of 3 mg/kg/day during 14 days presents positive effects on improving the physical performance tests related to lactic anaerobic metabolism, especially anaerobic power in soccer players.”
Line 52: Did you mean increase in glycogen replenishment rate?
Answer: Thank you for your proposal. The authors have changed recovery of muscle glycogen by increase in glycogen replenishment rate.
Line 54-56: Grammar check and it should be re-worded; and It doesn’t connect well with the previous statement.
Answer: Thank you for your comment. The authors have checked and reworded that sentence to connect with the previous statement. Thus, the new sentence is: “In this line, there are many sports modalities that its practice involves the combination of high intensity actions, in isolation or repeatedly, where is necessary a good anaerobic metabolism, with low intensity actions periods, where is necessary a good aerobic metabolism [8].”
Line 65 Bout should be a better word to use here.
Answer: Thank you for your correction. The authors have changed moment by bout.
Line 70: Doesn’t sound like an academic writing style!
Answer: Thank you for your comment. The authors have changed that sentence for sound like an academic writing style. The new sentence is: “Similar results have been observed when it talks about tests that use phosphagen metabolism.”
Line 76: What does this mean? Is it like the overall performance? More clarification is required here.
Answer: Thank you for your interest. The authors have added “soccer athletic performance” to clarify the meaning of the final effect.
Line 78: Does this mean overall soccer performance?
Answer: Thank you for your interest. In the same way that above answer, the authors have added “soccer athletic performance” to clarify the meaning of the overall soccer performance.
Line 79: This is unclear.
Answer: Thank you for your commented. In order to clarify, the authors have changed that sentence by: “Although Cr could improve soccer athletic performance, to the best of the authors knowledge there is not a clear consensus on what kind of soccer skills, and therefore of energy system involved, could be more effective.”
Line 82-83: I don’t think this is necessary.
Answer: Thank you for your suggestion. The authors have deleted that sentence.
Line 88: This should be consistent with the previous ones and throughout the paper.
Answer: In order for this to be consistent with the previous ones and throughout the paper, the authors have changed soccer physical performance by soccer athletic performance.
Line 118: Was creatine considered to be the main ingredient in theist products?
Answer: Given that all the studies analyzed consumed creatine directly and not through products that contained creatine, the authors have eliminated “or and Cr-containing product-“ in order not to generate confusion.
Line 128: type of condition? clinical condition?
Answer: Thank you for your suggestion. The authors have included “medical or illness condition or injury” in that sentence.
Line 196: How soccer can be different from soccer? This should be removed; perhaps an error here.
Answer: Thank you for your suggestion. The authors have changed soccer by American football.
Line 203-204: I would like to see the BMI and its changes during the course of studies as it might help with interpreting the results.
Answer: The authors appreciate the suggestion of the reviewer. However, these data are not shown in most studies, which makes it difficult to interpret the results under this prism. In this sense, the authors have included in the paragraph of limitations the following sentence: “Another limitation of this review is related to the impossibility of presenting body composition data that could help us interpret the results.”
Table2: Age is not mentioned here. Please add in Biwer et al., study.
Answer: Thank you for your suggestion. However, Biwer et al., do not present participant age. Therefore, the authors have included in the table “No age presented”.
Line 234: Was this repeated too?
Answer: Thank you for your comment. In this case the authors have deleted performance in “Wingate test performance” so that the same problem is not repeated.
Line 242-243: Why both “large a significant” are used here?
Answer: Thank you for your comment. The authors deleted large to avoid misunderstandings, since this word indicates the heterogeneity of the studies.
Line 245-246: Would you consider this as the final conclusion? I believe that we already knew Cr could be effective for soccer players but your main goal was to take into consideration different metabolic energy systems in this paper.
Answer: Thank you for your suggestion. The authors have added “for improve lactic anaerobic performance” in that sentence: “Thus, this meta-analysis suggests that Cr supplementation could be an ergogenic aid for improve lactic anaerobic performance in soccer players.”
Line 248: 3-5 days based on the literature.
Answer: Thank you for your suggestion. The authors have included “for 3-5 days” in that sentence.
Line 249: Reference is require here.
Answer: Thank you for your suggestion. The authors have added reference 31 in that sentence.
Line 260-261: I would recommend replacing this with exercise capacity.
Answer: Thank you for your suggestion. The authors have added “exercise capacity” in that sentence. The new sentence is: “Thus, the supplementation with Cr could lead to greater adaptations of training due to a higher quality and exercise capacity, as well as a better quicker recovery period [37].”
Line 261: Do you mean quicker recovery period?
Answer: Thank you for your suggestion. The authors have added “quicker recovery period” in that sentence. The new sentence is: “Thus, the supplementation with Cr could lead to greater adaptations of training due to a higher quality and exercise capacity, as well as a better quicker recovery period [37].”
Line 262: I don’t see any point of mentioning this here as it doesn’t align with the purpose of this review.
Answer: Thank you for pour comment. The author have deleted next sentence because it doesn’t align with the purpose of this review: “In terms of potential medical applications, Cr is intimately involved in several metabolic pathways [38].”
Line 266: How about the extra time? It’s probably worth mentioning it here.
Answer: Thank you for your suggestion. The authors have added “and sometimes 30 minutes of extra time in that sentence. The new sentence is: “Soccer requires a great aerobic capacity based on the duration of a match (90 minutes and sometimes 30 minutes of extra time) [40].”
Line 267: Please clarify what you mean by” level of equipment?
Answer: Thank you for your comment. The authors have changed “level of equipment” by “the competitive level of the team”.
Line 285-287: Wouldn’t this only be applicable to professional athletes?
Answer: Thanks for your suggestion. However, the authors do not consider that a Highly trained aerobic metabolism is exclusive to professional soccer players. In this sense, to become a professional player in addition to having a good physical development, including the aerobic, they must have other qualities such as technical-tactics, that a lower level player may lack despite having a highly trained aerobic metabolism.
Line 298: Did you mean PCr?
Answer: Thank you for your help. Effectively CP means PCr and the authors have changed that.
Line 326: I would recommend going in a bit of detail here as to how creatine may improve glycogen resynthesize.
Answer: Thank you for your interest. The authors have included a bit of detail here as to how creatine may improve glycogen resynthesize. The new sentence is: “This effect is thought to be the result of increased cell size due to Cr-induced water retention and is associated with the upregulation of signalling pathways mediating glycogen and protein synthesis, namely 5‟ AMP- activated protein kinase (AMPK)- and mechanistic target of rapamycin (mTOR)-mediated signaling [37].”
Line 348: Again, we need to be precise here in terms of the duration of Cr supplementation
Answer: Thank you for your recommendation. The authors have specified in terms of the duration of Cr supplementation. The new sentence is: “The effective dose of Cr supplementation to obtain positive effects describes with a load dose of 20-30 g / day divided 3-4 times a day ingested for 6-7 days followed by 5 g/day until 9 weeks or with a low dose of 3 mg/kg/day during 14 days or more.”

Round 2
Reviewer 2 Report
I would like to thank the authors for their responses as well as revisions you have made to improve your manuscript and appreciate taking my comments into consideration. Please see the attached file.
See the below to consider several language changes as well.
Line 18-20: The opening sentence is a bit unclear so I would suggest re-phrasing it.
Answer: Thank you for your suggestion. The authors have re-phrased these lines as: “Studies have shown that creatine supplementation increases intramuscular creatine concentrations, favoring the energy system of phosphagen, which may help explain the observed improvements in high intensity exercise performance. However, research on soccer physical performance has shown controversial results in part because the energy system used has not been taken into account.”
Line 27: What does that mean? Identical in terms of what? Taste, form, color
Answer: Thank you for your interest. The authors have included into brackets (the doses, duration, timing and drug appearance) to explain what means identical.
Line 38: I would recommend providing the exact time range here as thic could be misleading
Answer: Thank you for your recommendation. Taking into account that the improvements have been shown in 2 different protocols, the authors have re-written abstract conclusion as: “In conclusion, creatine supplementation with a high dose of 20-30 g divided 3-4 times/day ingested for 6-7 days followed by 5 g/day until 9 weeks or with a low dose of 3 mg/kg/day during 14 days presents positive effects on improving the physical performance tests related to lactic anaerobic metabolism, especially anaerobic power in soccer players.”
Line 52: Did you mean increase in glycogen replenishment rate?
Answer: Thank you for your proposal. The authors have changed recovery of muscle glycogen by increase in glycogen replenishment rate.
Line 54-56: Grammar check and it should be re-worded; and It doesn’t connect well with the previous statement.
Answer: Thank you for your comment. The authors have checked and reworded that sentence to connect with the previous statement. Thus, the new sentence is: “In this line, asmany team sports involves the combination of high intensity actions, in isolation or repeatedly, greater anaerobic and aerobic metabolism are necessary. [8].”
Line 65 Bout should be a better word to use here.
Answer: Thank you for your correction. The authors have changed moment by bout.
Line 70: Doesn’t sound like an academic writing style!
Answer: Thank you for your comment. The authors have changed that sentence for sound like an academic writing style. The new sentence is: “Similar results have been observed when it talks about tests that use phosphagen metabolism.”
Line 76: What does this mean? Is it like the overall performance? More clarification is required here.
Answer: Thank you for your interest. The authors have added “soccer athletic performance” to clarify the meaning of the final effect.
Line 78: Does this mean overall soccer performance?
Answer: Thank you for your interest. In the same way that above answer, the authors have added “soccer athletic performance” to clarify the meaning of the overall soccer performance.
Line 79: This is unclear.
Answer: Thank you for your commented. In order to clarify, the authors have changed that sentence by: “Although Cr could improve soccer athletic performance, to the best of the authors knowledge there is not a clear consensus on what kind of soccer skills, and therefore of energy system involved, could be more effective.”
Line 82-83: I don’t think this is necessary.
Answer: Thank you for your suggestion. The authors have deleted that sentence.
Line 88: This should be consistent with the previous ones and throughout the paper.
Answer: In order for this to be consistent with the previous ones and throughout the paper, the authors have changed soccer physical performance by soccer athletic performance.
Line 118: Was creatine considered to be the main ingredient in theist products?
Answer: Given that all the studies analyzed consumed creatine directly and not through products that contained creatine, the authors have eliminated “or and Cr-containing product-“ in order not to generate confusion.
Line 128: type of condition? clinical condition?
Answer: Thank you for your suggestion. The authors have included “medical condition or injury” in that sentence.
Line 196: How soccer can be different from soccer? This should be removed; perhaps an error here.
Answer: Thank you for your suggestion. The authors have changed soccer by American football.
Line 203-204: I would like to see the BMI and its changes during the course of studies as it might help with interpreting the results.
Answer: The authors appreciate the suggestion of the reviewer. However, these data are not shown in most studies, which makes it difficult to interpret the results under this prism. In this sense, the authors have included in the paragraph of limitations the following sentence: “Another limitation of this review is related to the impossibility of presenting body composition data that could help us interpret the results.”
Table2: Age is not mentioned here. Please add in Biwer et al., study.
Answer: Thank you for your suggestion. However, Biwer et al., do not present participant age. Therefore, the authors have included in the table “No age presented”.
Line 234: Was this repeated too?
Answer: Thank you for your comment. In this case the authors have deleted performance in “Wingate test performance” so that the same problem is not repeated.
Line 242-243: Why both “large a significant” are used here?
Answer: Thank you for your comment. The authors deleted large to avoid misunderstandings, since this word indicates the heterogeneity of the studies.
Line 245-246: Would you consider this as the final conclusion? I believe that we already knew Cr could be effective for soccer players but your main goal was to take into consideration different metabolic energy systems in this paper.
Answer: Thank you for your suggestion. The authors have added “for improve lactic anaerobic performance” in that sentence: “Thus, this meta-analysis suggests that Cr supplementation could be an ergogenic aid in order to improve lactic anaerobic performance in soccer players.”
Line 248: 3-5 days based on the literature.
Answer: Thank you for your suggestion. The authors have included “for 3-5 days” in that sentence.
Line 249: Reference is require here.
Answer: Thank you for your suggestion. The authors have added reference 31 in that sentence.
Line 260-261: I would recommend replacing this with exercise capacity.
Answer: Thank you for your suggestion. The authors have added “exercise capacity” in that sentence. The new sentence is: “Thus, the supplementation with Cr could lead to greater adaptations of training due to a higher quality and exercise capacity, as well as a quicker recovery period [37].”
Line 261: Do you mean quicker recovery period?
Answer: Thank you for your suggestion. The authors have added “quicker recovery period” in that sentence. The new sentence is: “Thus, the supplementation with Cr could lead to greater adaptations of training due to a higher quality and exercise capacity, as well as a quicker recovery period [37].”
Line 262: I don’t see any point of mentioning this here as it doesn’t align with the purpose of this review.
Answer: Thank you for pour comment. The author have deleted next sentence because it doesn’t align with the purpose of this review: “In terms of potential medical applications, Cr is intimately involved in several metabolic pathways [38].”
Line 266: How about the extra time? It’s probably worth mentioning it here.
Answer: Thank you for your suggestion. The authors have added “and sometimes 30 minutes of extra time in that sentence. The new sentence is: “Soccer requires a great aerobic capacity based on the duration of a match (90 minutes and sometimes 30 minutes of extra time) [40].”
Line 267: Please clarify what you mean by” level of equipment?
Answer: Thank you for your comment. The authors have changed “level of equipment” by “the competitive level of the team”.
Line 285-287: Wouldn’t this only be applicable to professional athletes?
Answer: Thanks for your suggestion. However, the authors do not consider that a Highly trained aerobic metabolism is exclusive to professional soccer players. In this sense, to become a professional player in addition to having a good physical development, including the aerobic, they must have other qualities such as technical-tactics, that a lower level player may lack despite having a highly trained aerobic metabolism.
Line 298: Did you mean PCr?
Answer: Thank you for your help. Effectively CP means PCr and the authors have changed that.
Line 326: I would recommend going in a bit of detail here as to how creatine may improve glycogen resynthesize.
Answer: Thank you for your interest. The authors have included a bit of detail here as to how creatine may improve glycogen resynthesize. The new sentence is: “This effect is thought to be the result of increased cell size due to Cr-induced water retention and is associated with the upregulation of signalling pathways mediating glycogen and protein synthesis, namely 5‟ AMP- activated protein kinase (AMPK)- and mechanistic target of rapamycin (mTOR)-mediated signaling [37].”
Line 348: Again, we need to be precise here in terms of the duration of Cr supplementation
Answer: Thank you for your recommendation. The authors have specified in terms of the duration of Cr supplementation. The new sentence is: “The effective dose of Cr supplementation to obtain positive effects describes with a loading dose of 20-30 g divided 3-4 times/day ingested for 6-7 days followed by 5 g/day until 9 weeks or with a low dose of 3 mg/kg/day during 14 days or more.”

Author Response
Dear reviewer:
We appreciate the effort you have made in helping us improve the article. In this sense, the authors have sent the manuscript to the English edition service of the MDPI publishing house to improve itsunderstanding. The authors hope that everything is correct.
